# GENERATING ANTIMICROBIAL PEPTIDES FROM LATENT SECONDARY STRUCTURE SPACE

## ABSTRACT

Antimicrobial peptides (AMPs) have shown promising results in broad-spectrum antibiotics and resistant infection treatments, making it a magnet for the field of drug discovery. Recently, an increasing number of researchers have been introducing deep generative models to AMP design. However, few studies consider secondary structure information during the generation, though it has shown crucial influence on antimicrobial activity in all AMP mechanism theories. This paper proposes LSSAMP which uses the multi-scale VQ-VAE to learn the positional latent spaces modeling the secondary structure. By Sampling in the latent secondary structure space, we can generate peptides with ideal amino acids and secondary structures simultaneously. Experimental results show that our LSSAMP can generate peptides with multiple ideal physical attributes and a high probability of being predicted as AMPs by public AMP prediction models.

## 1 INTRODUCTION

Developing neural networks for drug discovery has attracted increasing attention recently. It can facilitate the discovery of potential therapies and reduce the time and cost of drug development (Stokes et al., 2020). Plenty of works have been done to employ deep generative models in searching for drug-like molecules with desired properties and achieved great success (Jin et al., 2018; Shi et al., 2019; Schwalbe-Koda & Gómez-Bombarelli, 2020; Xie et al., 2020). However, these works mainly focus on small molecules, and more complicated biochemicals, such as proteins, are still rarely explored.

Antimicrobial peptides (AMPs), defined as short proteins of less than 50 amino acids with potent antimicrobial activity, are an emerging category of therapeutic agents. AMPs exist widely in the natural immune system for all species and kill bacteria in a physical way (Aronica et al., 2021; Cardoso et al., 2020). They attach to the bacterial membrane and insert into the membrane to form pores, which leads to the death of bacteria by allowing cytoplasmic leakage. This mechanism makes them more promising for handling extensively drug-resistant bacteria than traditional antibiotics (Mahlapuu et al., 2016). However, the theoretical chemical space of peptides is enormous and the sequence number grows exponentially as the length increases. Thus, it is challenging to search for valid peptides with antimicrobial properties from such a huge sequence space.

Several factors can affect the antimicrobial activity of peptides (Boman, 2003). Amino acids with positive charges are more likely to bind with bacterial membrane as most bacterial surfaces are anionic, while those with high hydrophobicities tend to move from the solution environment to the bacterial membrane. However, the mechanisms of antibacterial peptides need not only a reasonable sequence but also an appropriate structure. For example, by forming the helix structure, a peptide can gather the hydrophobic amino acids on one side and hydrophilic ones on the other. This ability, named amphipathy, helps it insert into the membrane and maintain a stable hole with other peptide molecules in the membrane, as shown in Figure 1. The hole will drain the cytoplasm and finally kill the bacteria. This mechanism of killing the bacteria is called 'barrel stave'. Amphipathy plays an important role in deciding the antibacterial activity of peptides and is closely related to the secondary structure of the peptide (Aronica et al., 2021).

According to the antimicrobial mechanism, a new AMP should meet the following criteria. *C1*: It possesses several ideal physical attributes (e.g. positive charge and high hydrophobicity). *C2*: It

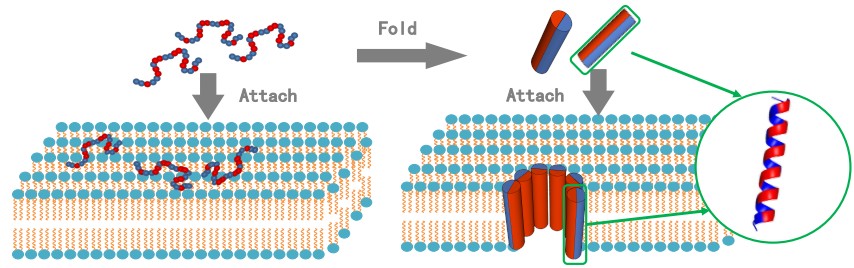

Figure 1: An example of the 'barrel stave' antimicrobial mechanism. The blue indicates the hydrophobic amino acids, and the red ones are hydrophilic. On the left, although the peptides with reasonable amino acids has attached to the bacterial membrane, they still can not insert into it. However, by folding into the helix structure, as shown on the right, the peptides maintain a stable hole which breaks the membrane of the bacterium.

has appropriate secondary structures (e.g. alpha-helix). *C3*: It differs from existing AMPs to some extent.

The existing works mainly focus on sequential features of amino acids and ignore the secondary structure. The traditional methods replace subsequences with patterns from the pattern database in a given template (Porto et al., 2018). Inspired by success in deep neural networks, many researchers apply neural generative models to AMP discovery. They often use the physical attributes as the extra input to control the generation phase (Das et al., 2018; Van Oort et al., 2021), or train classifiers on each attribute to filter the peptides after the generation (Capecchi et al., 2021; Das et al., 2021). The former ones usually generate peptides that have low correlation with the input attributes and the filter phase of the latter ones make the sampling inefficient.

As described above, the antimicrobial activity is determined by both the amino acid composition and secondary structure of the peptide. Thus, we propose LSSAMP to generate antimicrobial peptides from the latent semantic and structure space. Taking the peptide sequence as the time series, we assign a latent variable on each position. Since it is computationally intractable to sum continuous latent variables over all positions, we employ the vector quantized-variational autoencoder (VQ-VAE) (van den Oord et al., 2017) to learn the discrete distribution for each position and further design a multi-scale codebooks strategy to capture different local patterns to fit various length ranges for amino acid and structure sequences. During the generation process, LSSAMP will sample a backbone from the secondary structure latent space and generate the amino acid sequence simultaneously. We evaluate LSSAMP and several baselines through physical properties that are closely related to the antibacterial mechanism. Besides, we use some public AMP prediction models to predict generated sequences being AMPs as previous works did (Das et al., 2020; Van Oort et al., 2021).

To conclude, our contributions are as follows:

• We propose LSSAMP, a generative model which samples peptides from the latent secondary structure space to control the peptide properties.
• We develop a multi-scale VQ-VAE to learn positional latent spaces from different aspects and model semantic sequences and structural sequences in the same space.
• Experimental results show that LSSAMP can generate peptides with multiple ideal features such as positive charge, better hydrophobicity, and better amphipathicity. The results of public AMP classifiers also verify that our model can generate peptides with high AMP probability.

## 2 RELATED WORK

**Antimicrobial Peptides Generation** Traditional methods for AMP design can be divided into three approaches (Torres & de la Fuente-Nunez, 2019): (i) The pattern recognition algorithms build a sequential pattern database from existing AMPs, and then pick a template peptide and replace local sequence with patterns (Loose et al., 2006; Porto et al., 2018). (ii) The genetic algorithms analyze the AMP database and design some antibiotic activity functions (Maccari et al., 2013). (iii)The

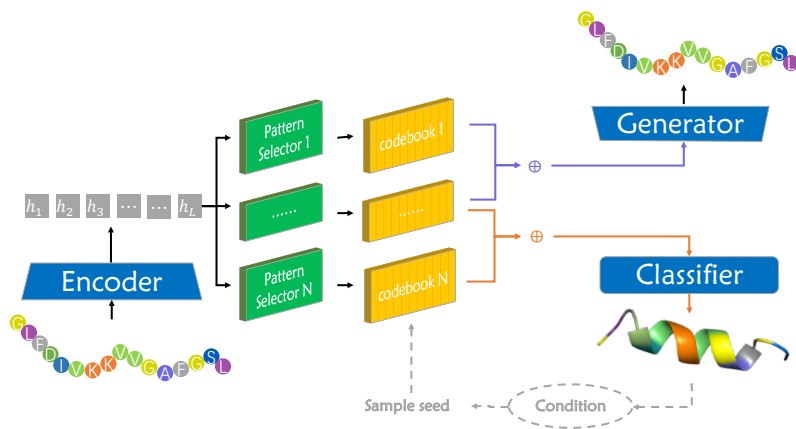

Figure 2: The framework of LSSAMP. The input sequence is first fed into the encoder to get the hidden representation $\boldsymbol{h}$. Then, we use $N$ pattern selectors to select local patterns with different scales and use the corresponding codebooks to obtain discrete latent variables for each position. The codebooks are made to learn the latent space of the secondary structure by employing the secondary structure prediction task on them. For inference, we generate peptides from the latent structure space by sampling the index sequences on the codebooks. We also try to use some conditions to restrict the structure motifs to further improve the generation performance.

molecular modeling and molecular dynamics methods build 3D models of peptides and analyze activity (Matyus et al., 2007; Bolintineanu & Kaznessis, 2011). Deep generative models take a rapid growth in recent years. Dean & Walper (2020) encodes the peptide into the latent space and interpolates across a predictive vector between a known AMP and its scrambled version to generate novel peptides. The PepCVAE (Das et al., 2018) and CLaSS (Das et al., 2021) employ the variational auto-encoder model to generate sequences. The AMPGAN (Van Oort et al., 2021) uses the generative adversarial network to generate new peptide sequences with conditions. To our knowledge, this is the first study to take secondary structure information into consideration during the generative phase, which is conducive to effectively generate well-structured sequences with desired properties .

**Sequence Generation via VQ-VAE** The variational auto-encoders (VAEs) were first proposed by Kingma & Welling (2014) for image generation, and then widely applied to sequence generation tasks such as language modeling (Bowman et al., 2016), paraphrase generation (Gupta et al., 2018), machine translation (Bao et al., 2019) and so on. Instead of mapping the input to a continuous latent space in VAE, the vector quantized-variational autoencoder (VQ-VAE) (van den Oord et al., 2017) learns the codebook to obtain a discrete latent representation. It can avoid issues of posterior collapse while has comparable performance with VAEs. Based on it, Razavi et al. (2019) uses a multi-scale hierarchical organization to capture global and local features for image generation. Bao et al. (2021) learns implicit categorical information of target words with VQ-VAE and models the categorical sequence with conditional random fields in non-autoregressive machine translation. In this paper, we employ the multi-scale vector quantized technique to obtain the discrete representation for each position of the peptide.

## 3 METHOD

Given a peptide sequence $\boldsymbol{x} = \{a_1, a_2, \cdots, a_L\}$, where $a$ belongs to the 20 common amino acids and $L$ is the sequence length, the corresponding secondary structure can be denoted as $\boldsymbol{y} = \{y_1, y_2, \cdots, y_L\}$. Following the definition in Kabsch & Sander (1983), there are 8 secondary structure types, including one unknown label, so $y_i \in \{H, B, E, G, I, T, S, -\}$[1]. We first employ VQ-VAE for the sequence reconstruction task to learn the sequential latent space (Section 3.1). Then, we enforce the latent space to model the structure information by the secondary structure task (Section 3.2). Besides, we design the multi-scale codebooks to capture different local patterns (Sec-

---

[1] $H, G, I$ denote the alpha, 3-10, and pi helix. $E, T$ are the strand and turn. The others are coil structures.

tion 3.3). Finally, we describe the training and inference phase in Section 3.4. The overview of our model is shown in Figure 2.

## 3.1 Modeling Peptide Sequences

For sequential information, we embed the input peptide $x = \{a_1, a_2, \cdots, a_L\}$ to the latent space via the encoder and use the generator to reconstruct $x$. We assume that each $a_i$ is determined by a latent variable $z_i$, and the input sequences $x = a_{1:L}$ will be assigned to a latent sequence $z = z_{1:L}$. Since it is computationally intractable to sum continuous latent variables over the sequence, we use VQ-VAE (van den Oord et al., 2017) to lookup the discrete embedding vector $z_q = \{z_q(a_1), \cdots, z_q(a_L)\}$ for each position by vector quantization.

Specifically, the encoder output $z_e(a_i) \in \mathbb{R}^d$ will be replaced by the codebook embedding $z_q(a_i) \in \mathbb{R}^d$ via a nearest neighbors lookup from the codebook $B \in \mathbb{R}^{K \times d}$ :

$$z_q(a_i) = e_k, \text{and } k = \mathrm{argmin}_{j \in \{1, \cdots, K\}} \|z_e(a_i) - e_j\|_2 . \tag{1}$$

Here, $K$ is the size of the codebook and $d$ is the dimension of the codebook entry $e$. Then, the generator will take $z_q(a_i)$ as its input and reconstruct $x$. The training objective $L_r$ is defined as:

$$L_r = \log p\left(a_i | z_q(a_i)\right) + \|\mathrm{sg}\left[z_e(a_i)\right] - z_q(a_i)\|_2^2 + \beta \|z_e(a_i) - \mathrm{sg}[z_q(a_i)]\|_2^2 . \tag{2}$$

Here, $sg(\cdot)$ is the stop gradient operator, which becomes 0 at the backward pass. $\beta$ is the commit coefficient to control the codebook loss.

## 3.2 Modeling Secondary Structures

In order to model the categorical information of the secondary structure, we define an 8-category sequence labeling task on the latent space, which takes $x$ as the input and the structure label sequence $y$ as the target. Similar with sequence reconstruction, we use the same encoder to get $z_e(a_i)$ and employ VQ-VAE to obtain discrete representation. Then, $z_q'(a_i)$ is fed to a separate classifier for the secondary structure prediction:

$$L_s = \log p\left(y_i | z_q'(a_i)\right) + \left\|\mathrm{sg}\left[z_e(a_i)\right] - z_q'(a_i)\right\|_2^2 + \beta \left\|z_e(a_i) - \mathrm{sg}[z_q'(a_i)]\right\|_2^2 . \tag{3}$$

Peptide sequences and structures have distinctive local features, which are often utilized in traditional design algorithms. The patterns of amino acids are often used for template-based design and feature-based recognition. For structure motifs such as $\alpha$-helix with at least 3.6 consecutive amino acids, they will determine the position of amino acids in the 3D space and affect the function of peptides. However, the structure motifs are often much longer than sequence patterns. Therefore, we establish codebooks of multiple scales to learn latent spaces for different local patterns.

## 3.3 Multi-scale VQ-VAE

For the encoder output $h_i = z_e^{(0)}(a_i)$, we first use $N$ pattern selectors $F^{(n)}$ to extract local patterns from different scales, which will get $z_e^{(n)}(a_i) = F^{(n)}(h_i)$. Then, we establish the codebook for each pattern, and use Eqn. 1 to look up the nearest codebook embedding $z_q^{(n)}(a_i)$.

We share the codebooks between the sequence reconstruction and secondary structure prediction to capture common features and relationships between the amino acid and its structure. Specifically, we make $N_r$ codebook for the sequence information and $N_s$ for the structure. The $z_{q_r}(a_i)$ is the concatenation of $N_r$ codebook embeddings and is fed to the sequence generator:

$$z_{q_r}(a_i) = \|_{n \in N_r} z_q^{(n)}(a_i), \tag{4}$$

Based on Eqn. 2, the reconstruction training objective for multi-scale VQ-VAE can be denoted as:

$$L_r = \log p\left(a_i | z_{q_r}(a_i)\right) + \sum_{n \in N_r} \left(\left\|\mathrm{sg}\left[z_e^{(n)}(a_i)\right] - z_q^{(n)}(a_i)\right\|_2^2 + \beta \left\|z_e^{(n)}(a_i) - \mathrm{sg}[z_q^{(n)}(a_i)]\right\|_2^2\right). \tag{5}$$

---

**Algorithm 1** Training phase of `LSSAMP`

---

**Require:** A pre-training dataset $D_r$, a peptide dataset with secondary structure $D_s$, and the AMP dataset $D_{amp}$. The model $M_\theta$.
1: Pre-train sequence reconstruction on $D_r$ and update $M_\theta$ via Eqn. 5.
2: Finetune sequence reconstruction as well as secondary structure prediction on $D_s$ and update the $M_\theta$ via Eqn. 6.
3: Further Finetune $M_\theta$ on $D_{amp}$.
4: **for** $n = 1, 2, \cdots, N$ **do**
5:     Create an empty dataset $C_n$.
6:     **for** $x_i \in D_{amp}$ **do**
7:         Save the $n$-th codebook index of $x_i$ via Eqn. 1 to $C_n$
8:     **end for**
9:     Train an auto-regressive language model $M_{prior_n}$ on $C_n$.
10: **end for**

---

The loss of secondary structure prediction task $L_s$ can be formulated in the same way from Eqn. 3. And the total training loss is:

$$L = L_r + \gamma L_s, \tag{6}$$

where the $\gamma$ is the weight of the secondary structure prediction.

### 3.4 TRAINING AND INFERENCE

We discuss the training and sampling process in this section. Since the AMP dataset $D_{amp}$ is very small, we use two extra datasets $D_r$ and $D_s$ to pre-train the reconstruction and prediction task. The whole training process is described in Algorithm 1.

**Training** To enable the generator to generate valid peptide sequences, we first pre-train our model on a large protein dataset $D_r$ for the sequence construction task. Then, we jointly train the sequence reconstruction and the structure prediction on a peptide dataset $D_s$ with secondary structure information. This phase maps the positional latent space to an entangled sequential and structural distribution. Finally, we finetune our model on the small AMP dataset $D_{amp}$ to capture the characteristics of AMPs.

Following Kaiser et al. (2018), we use Exponential Moving Average (EMA) to update the embedding vectors in the codebooks. Specifically, we keep a count $c_k$ measuring the number of times that the embedding vector $e_k$ is chosen as the nearest neighbor of $z_e(a_i)$ via Eqn. 1. Thus, the counts are updated with a sort of momentum: $c_k \leftarrow \lambda c_k + (1 - \lambda) \sum_i \mathbb{I}[z_q(a_i) = e_k]$, with the embedding $e_k$ being updated as: $e_k \leftarrow \lambda e_k + (1 - \lambda) \sum_i \frac{\mathbb{I}[z_q(a_i) = e_k] z_e(a_i)}{c_k}$. Here, $\lambda$ is the decay parameter.

**Prior Model** The prior distribution over the codebook is a categorical distribution and can be made auto-regressive by the extra prior model. In order to model the dependency between $z_{1:L}$, we train Transformer-based language models on the embedding entries. We extract the index sequences generated by Eqn. 1 for each codebook $n$ and then train $M_{prior_n}$ on them.

**Inference** We sample several index sequences from the prior models for each codebook $n$, and then lookup the codebook to get the embedding vector $\boldsymbol{z}_q^{(n)}$. Finally, $\boldsymbol{z}_q^{(n)}$ is fed to the generator and classifier to generate the sequence with its secondary structure. We also try to control the secondary structure by existing AMP structure patterns to further improve the generation quality.

## 4 EXPERIMENT

### 4.1 EXPERIMENT SETUP

**Dataset** The Universal Protein Resource (UniProt)[2] is a comprehensive protein dataset. We download reviewed protein sequences (550k) with the limitation of 100 in length as $D_r$ (57k examples). Then we use ProSPr[3] (Billings et al., 2019) to predict the secondary structure for $D_r$ and filter some

---

[2]https://www.uniprot.org/
[3]https://github.com/dellacortelab/prospr/tree/prospr1

low-quality examples. Therefore, we get $D_s$ with 46k examples, which have both sequence and secondary structure information. For antimicrobial peptide dataset, we download from Antimicrobial Peptide Database (APD3)[4] (Wang et al., 2016) and filter repeated ones to get 3222 AMPs as $D_{amp}$. We randomly extract 3,000 examples as validation and 3,000 as test on $D_r$ and $D_s$. For $D_{amp}$, the size of validation and test is both 100. Following Veltri et al. (2018), we establish a decoy set with negative examples that do not have antimicrobial activities to examine the evaluation metric. It first removes peptide sequences with anti-microbe activity from Uniprot, and then removes sequences with length $< 10$ and sequences with $> 40$ sequence identity with AMP sequences, resulting in 2021 non-AMP sequences.

**Baseline** We implement several baselines for our experiment. Traditional methods usually randomly substitute several amino acids on the existing AMPs and conducting biological experiments on them. Thus, we use the **Random** replacement with probability $p$ as the baseline. Following Dean & Walper (2020), we use **VAE** to embed the peptides into the latent space and sample latent variable $z$ from the standard Gaussian distribution $p \sim N(0, 1)$. For a fair comparison, we use the same Transformer architecture as our model `LSSAMP` and train on the Uniprot $D_r$ and APD dataset $D_{amp}$. **AMP-GAN** is proposed by Van Oort et al. (2021), which uses a BiCGAN architecture with convolution layers. It consists of three components: the generator, discriminator, and encoder. The generator and discriminator share the same encoder. It is trained on 49k false negative sequences from UniProt and 7k positive AMP sequences. **PepCVAE** (Das et al., 2018) is a VAE-based generative model with a structured variable $c$ to control the attribute of the sequence. Since the authors did not release their code, we use the model architecture from Hu et al. (2017) and modify the reproduced code[5] for AMPs. The original paper use 93k sequences from UniProt and 7960/6948 positive/negative AMPs for training. For comparison, we use our UniProt dataset $D_r$ and ADP dataset $D_{amp}$ to train it. **MLPeptide** (Capecchi et al., 2021) is RNN-based generator. It is first trained on 3580 AMPs and then transfer against specific bacteria.

**Implementation Details** There are three main modules for `LSSAMP`. The encoder and decoder are based on 2-layer Transformer (Vaswani et al., 2017) with $d_{model} = 128$ and $head = 8$. The size of FFN projection is $d_{ffn} = 512$ and all drouput rate are $0.1$. For the classifier, we use the same CNN block as Billings et al. (2019) with 32 input channels and a dilation scale of $[1, 2, 4, 8, 10]$. For multi-scale codebooks, we first apply CNN as $F^{(n)}$ to extract features. We set $n = 4$ and kernel width ranging in $[1, 2, 4, 8]$. The features will be padded to the same length as the input sequence. Then, we use 4 codebooks with $K = 8$ and $d = 128$. The reconstruction and prediction share the same codebooks, which means $N_r = N_s = 4$. The commit coefficient is set to $\beta = 0.05$.

We use PyTorch for implementation and train on the single Tesla-V100-32GB. We optimize the parameter with Adam Optimizer (Kingma & Ba, 2015). During pre-training for sequence construction on $D_r$, we limit the max length to 100 and set the maximum tokens in a batch $bz$ as 30,000, learning rate $lr$ as 0.01 with 8,000 warmup steps, and decoy weight for EMA as $\lambda = 0.8$. For secondary structure prediction on $D_s$, the max length is limited to 32, $bz = 10,000$, $lr = 0.003$, $\lambda = 0.95$, and the prediction loss coefficient $\gamma = 1$. Finally, we transfer to $D_{amp}$ with the same hyperparameters except the $lr = 0.001$.

## 4.2 EVALUATION METRIC

To evaluate the antimicrobial activity of the generated peptide $S = \{a_1, a_2, \cdots, a_L\}$, we use several physical features according to the mechanism of AMPs. Besides, we also apply some open-source AMP predictive models to estimate the probability of the generated peptides being AMPs.

### 4.2.1 PHYSICAL ATTRIBUTES

**Charge** The bacterial membrane usually takes the negative charge, so peptides with the positive charge prefer to bind with the bacteria. The whole net charge of the peptide sequence $S$ is defined as the sum of the charge of all its amino acids $C(a_i)$ at pH 7.4, which is $C(S) = \sum_{a_i \in S} C(a_i)$
**Hydrophobicity** The hydrophobicity reflects the tendency to bind lipids on the bacterial membrane. It is more likely for the peptide with a high hydrophobicity to move from the solution environment

---

[4]https://aps.unmc.edu/
[5]https://github.com/wiseodd/controlled-text-generation

to the bacterial membrane. We use the hydrophobicity scale $H(a_i)$ in Eisenberg et al. (1984) to calculate the hydrophobicity of a sequence, which is $H(S) = \sum_{a_i \in S} H(a_i)$

**Amphipathicity / Hydrophobic Momentum** The amphipathicity measures the ability of the peptide to bind water and lipid at the same time, which is a definitive feature of antimicrobial peptides(Hancock & Rozek, 2002). It can be quantified by the hydrophobic momentum $uH(S, \theta)$, defined by Eisenberg et al. (1984). The hydrophobic momentum is determined by the hydrophobicity $H(a_i)$ of each amino acid $a_i$, along with the angle $\theta$ between amino acids. The angle can be estimated by the secondary structure. For the $\alpha$-helix structure, $\theta$ is $100°$ and for $\beta$-sheet, $\theta$ is $180°$.

$$uH(S, \theta) = \sqrt{\left(\sum_{a_i \in S} H(a_i) * cos(i * \theta)\right)^2 + \left(\sum_{a_i \in S} H(a_i) sin(i * \theta)\right)^2} \tag{7}$$

For each peptide, we calculate the above properties to measure its antimicrobial activity. For comparison, we draw the distribution on APD and decoy dataset and choose a range for each property[6]. We use the percentage of peptides in the range for each attribute to leverage the generation performance and use **Combination** to measure the percentage of those that satisfy three conditions at the same time. The detailed information is attached in the appendix.

| | Uniq | C | H | uH | Combination |
|---|---|---|---|---|---|
| **APD** | 3222 | 68.75% | 27.96% | 4.72% | 6.15% |
| **Decoy** | 2020 | 21.83% | 8.81% | 1.98% | 0.10% |
| **Random** $p = 0.1$ | 4978 | 66.16% $\pm$ 0.21% | 26.92% $\pm$ 0.24% | 23.12% $\pm$ 0.58% | 4.40% $\pm$ 0.16% |
| **Random** $p = 0.2$ | 5000 | 61.70% $\pm$ 0.39% | 24.87% $\pm$ 0.29% | 20.79% $\pm$ 0.76% | 2.47% $\pm$ 0.17% |
| **VAE** | 4988 | 38.54% $\pm$ 0.36% | 21.37% $\pm$ 0.58% | 12.60% $\pm$ 0.67% | 0.34% $\pm$ 0.11% |
| **AMP-GAN** | 4976 | **88.07**% $\pm$ 0.42% | 17.39% $\pm$ 0.75% | 23.55% $\pm$ 0.72% | 1.93% $\pm$ 0.05% |
| **PepCVAE** | 1346 | 58.89% $\pm$ 1.05% | 14.54% $\pm$ 0.55% | 11.65% $\pm$ 0.23% | 2.75% $\pm$ 0.25% |
| **MLPeptide** | 4486 | 86.59% $\pm$ 0.55% | 9.01% $\pm$ 0.33% | **36.56**% $\pm$ 0.62% | 3.22% $\pm$ 0.19% |
| **LSSAMP** | 4886 | 75.06% $\pm$ 0.37% | **39.51**% $\pm$ 0.64% | 27.18% $\pm$ 0.41% | **9.96**% $\pm$ 0.07% |
| **LSSAMP w/o condition** | 4893 | 60.88% $\pm$ 0.44% | 38.75% $\pm$ 0.26% | 24.05% $\pm$ 0.71% | 6.42% $\pm$ 0.29% |

Table 1: Physical attributes of generated sequences. We use the percentage of peptides meeting the range to measure the performance. **Uniq** is the number of unique generated sequences. **C**, **H**, **uH** correspond to charge, hydrophobicity, hydrophobic moment described in Section 4.2.1. **Combination** is the percentage satisfying three ranges at the same time. The best results are bold.

### 4.2.2 AMP Classifiers

Following previous works (Das et al., 2020; Van Oort et al., 2021), we also use several open-source AMP prediction tools to predict the AMP probability of generated sequence. Since these prediction tools are trained and report results in different AMP datasets, we first use them to predict sequences in APD and decoy dataset as a reference of their performance.

Thomas et al. (2010) trained on the AMP database of 3782 sequences with random forest (**RF**), discriminant analysis (**DA**), support vector machines (**SVM**)[7], and artificial neural network (ANN)[8] respectively. AMP Scanner v2[9] (Veltri et al., 2018), short as **Scanner**, is a CNN-&LSTM-based deep neural network trained on 1778 AMPs picked from APD. **AMPMIC**[10] (Witten & Witten, 2019) trained a CNN-based regression model on 6760 unique sequences and 51345 MIC measurement to predict MIC values. **IAMPE**[11] (Kavousi et al., 2020) is a model based on Xtreme Gradient Boosting. It achieves the highest correct prediction rate on a a set of ten more recent AMPs (Aronica et al., 2021). **ampPEP**[12] (Lawrence et al., 2021) is a random forest based model which is trained on 3268 AMP sequences. It has the best performance across multiple data sets (Aronica et al., 2021).

---

[6]The requirements are $C \in [2, 10]$, $H \in [0.25, \infty]$, and $uH \in [0.5, 0.75] \cup [1.75, \infty]$.

[7]http://www.camp3.bicnirrh.res.in/prediction.php

[8]We drop the ANN model becasue its accuracy on APD is low (82.83%).

[9]https://www.dveltri.com/ascan/v2/ascan.html

[10]https://github.com/zswitten/Antimicrobial-Peptides

[11]http://cbb1.ut.ac.ir/AMPClassifier/Index

[12]https://github.com/tlawrence3/amPEPpy

## 4.3 RESULTS AND ANALYSIS

We generate 5000 sequences for each baseline and filter the repeated ones. During the generation, we add some structural restrictions based on the antimicrobial mechanism on the secondary structure to improve the performance. Specifically, we reject peptides with more than 30% coil structure ('-'), which can hardly fold in the solution environment and insert into the bacterial membrane. Besides, we limit the minimum length of a helix structure ('H') to 4 based on physical rules. We name our model with structural control as **LSSAMP** and the model without extra conditions as **LSSAMP w/o condition**. We discuss the structure conditions and give some generated peptides in the appendix.

| | SVM | RF | DA | Scanner | AMPMIC | IAMPE | amPEP | Average |
|---|---|---|---|---|---|---|---|---|
| **APD** | 87.78% | 91.24% | 86.24% | 94.66% | 98.42% | 97.83% | 91.50% | 92.52% |
| **Decoy** | 17.43% | 13.71% | 16.04% | 0.25% | 18.07% | 23.53% | 52.92% | 20.28% |
| **Random** $p = 0.1$ | 86.06% | 86.12% | 84.01% | 93.23% | 79.14% | 95.60% | 91.74% | 87.99% |
| **Random** $p = 0.2$ | 76.66% | 76.64% | 74.83% | 86.95% | 68.57% | 91.14% | 87.89% | 80.38% |
| **VAE** | 24.90% | 15.30% | 13.83% | 15.12% | 15.25% | 40.31% | 24.30% | 21.29% |
| **AMP-GAN** | 78.62% | 87.29% | 83.82% | 82.17% | 89.58% | 93.88% | 80.52% | 85.13% |
| **PepCVAE** | 82.84% | 85.96% | **93.33%** | 85.44% | **98.44%** | **98.14%** | 80.77% | 89.27% |
| **MLPeptide** | 90.43% | **92.55%** | 93.08% | **93.72%** | 96.34% | 97.05% | 91.37% | **93.51%** |
| **LSSAMP** | **91.78%** | 89.86% | 91.09% | 90.51% | 93.40% | 90.42% | **93.23%** | 91.47% |
| **LSSAMP w/o condition** | 77.98% | 78.31% | 78.00% | 85.72% | 76.71% | 90.05% | 84.32% | 81.58% |

Table 2: The percentage of generated sequences being predicted as AMP. The classifiers are described in Section 4.2.2. The first part is the prediction results on AMP and non-AMP dataset as the reference. The bold ones are the best model results.

**Physical Attributes** As listed in Table 1, LSSAMP outperforms on the combination percentage by a large margin (5.56%), which indicates that our model can generate sequences satisfying multiple properties at the same time. If we do not control the structure, the combination percentage is similar to APD, which implies our model learns the latent distribution of APD. For the specific attribute, LSSAMP tends to generate peptides with higher hydrophobicity, while AMP-GAN and MLPeptide have more positive sequences. Compared with other models, PepCVAE tends to generate redundant sequences, which results in a significant decline in the number of unique sequences. Also, we can find that by control the secondary structure, all metrics can be improved. This verifies that secondary structure information has a great influence on peptide properties and it is beneficial to take it into consideration during the peptide generation.

**AMP Prediction** The results of prediction tools are shown in Table 2. By the comparison of LSSAMP and LSSAMP w/o condition, we can draw the same conclusion as above that controlling the secondary structure can further improve the generation performance. For AMPMIC and IAMPE, PepCVAE has a significant advantage over other methods, but it has a poor performance in other classifiers. LSSAMP performs better than MLPeptide in SVM and amPEP but does not perform as well as it in other classifiers. On the classifier with the largest performance gap IAMPE, we can find that the results of LSSAMP and LSSAMP w/o condition are similar, which implies that different secondary structures do not affect the results of IAMPE. However, in real scenarios, the structure is very important for the antimicrobial mechanism, which makes the classifier less reliable.

| | PPL ↓ | Loss ↓ | AA Acc.↑ | SS Acc.↑ |
|---|---|---|---|---|
| **LSSAMP** | **3.12** | **1.14** | **99.93** | **86.76** |
| w/o $D_r$ | 11.56 | 2.45 | 66.06 | 82.78 |
| w/o $D_s$ | 3.83 | 1.34 | 99.58 | 85.87 |
| w/o subbook | 3.49 | 1.25 | 99.86 | 86.61 |

Table 3: Ablation Study on validation set of $D_{amp}$. 'w/o' means that we remove the module from LSSAMP. ↓ means lower is better, while ↑ means higher is better. The detailed descriptions are in Section 4.3.

| Codebook | PPL ↓ | Loss ↓ | AA Acc.↑ | SS Acc.↑ |
|---|---|---|---|---|
| $[1]$ | 19.04 | 2.94 | 65.49 | 83.41 |
| $[1, 2]$ | 3.84 | 1.35 | 99.40 | 85.39 |
| $[1, 2, 4]$ | 3.32 | 1.20 | **100.00** | 85.95 |
| $[1, 2, 4, 8]$ | **3.24** | **1.17** | 99.79 | **87.20** |

Table 4: The influence of the number of codebooks. '[1,2,4,8]' indicates that we use 4 codebooks to capture local features with window sizes of 1,2,4,8. The meanings of symbols are the same as Table 3.

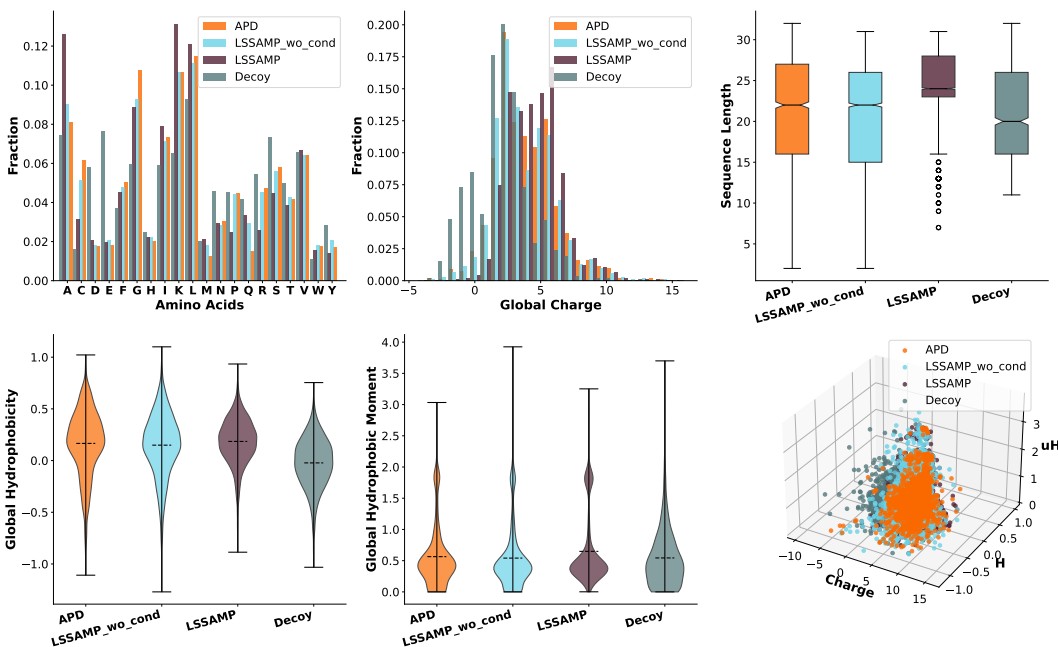

Figure 3: The distribution of amino acids, charge, sequence length, hydrophobicity, hydrophobic momentum, and a 3D visualization for three physical attributes.

**Ablation Study** We conduct the ablation study for our LSSAMP and show the results in Table 3. **PPL** is the perplexity of generated sequences that can measure fluency. **Loss** is the model loss on the validation set. **AA Acc.** is the reconstruction accuracy of amino acids sequences and **SS Acc.** is the prediction accuracy of the secondary structure. We can find that without the first pre-training phase on $D_r$ for sequence reconstruction, the model can hardly generate valid sequences. The second phase to train the model on the large-scale secondary structure dataset $D_s$ will affect the prediction performance on the target AMP dataset. If we remove multiple sub-codebooks and use a single codebook with the same size, the performance will decline a little.

**Codebook Number** We explore the effect of different numbers of codebooks on generation performance. From Table 4, we find that a single small codebook can hardly learn enough information to reconstruct the sequences. The PPL, loss, and secondary structure accuracy become better with the increase of codebook items. However, the reconstruction accuracy achieves the best performance when the codebook number is 3. This may be due to the relatively short local patterns for the amino acid sequence, which makes the window of size 8 too long for it.

**Visualization** We plot the distribution of amino acids, charge, sequence length, hydrophobicity, and hydrophobic momentum for APD, Decoy, and our models in Figure 3. Without condition, the distribution of LSSAMP is similar to APD, which indicates that LSSAMP successfully learns the latent space of APD. However, if we control the secondary structure, it is more likely to generate sequences with a longer length and more positive charges. For hydrophobicity and hydrophobic momentum, the distribution of generated sequences is more concentrated.

## 5 CONCLUSION

In this paper, we propose a multi-scale VQ-VAE to generate peptides. Motivated by the important role of structure in the antimicrobial mechanism, LSSAMP learns the latent spaces for each position with both sequential and structural features. Our model shows excellent performance on physical attributes related to antimicrobial activities and has a high probability to be predicted as AMPs by public classifiers. As the structural feature determines the function of macromolecules, LSSAMP has great potential for macromolecules design, including proteins and RNAs.

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

# A APPENDIX

## A.1 ATTRIBUTE DISTRIBUTION

To determine the threshold of net charge, hydrophobicity, and hydrophobic moment, we analyze the distribution of sequences in APD and decoy in Figure 4. For charge, we follow the rule summarized by specialists and choose sequences whose net charge is +2 to +10. For two characters left, we plot the histogram and compare the proportion in every bin. If the proportion of APD is larger than that in decoy, we add the bin into the acceptation range of the evaluation metric. The final ranges are $C \in [2, 10]$, $H \in [0.25, \infty]$, and $uH \in [0.5, 0.75] \cup [1.75, \infty]$.

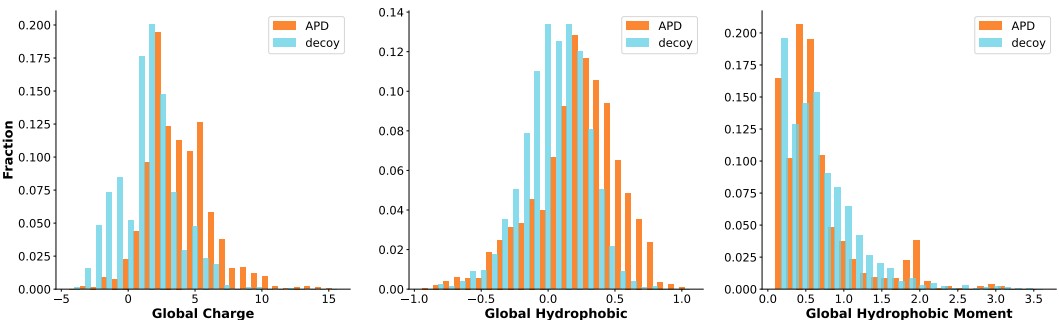

Figure 4: The histogram of charge, hydrophobicity and hydrophobic moment on APD and decoy dataset.

## A.2 GENERATED AMPS

We show 10 peptides generated by `LSSAMP` in Table 5, and build 3D models for four of them by PEPFold 3 (Shen et al., 2014) and draw the picture by PyMOL (Schrödinger, LLC, 2015c;a;b) in Figure 5. We can find that all these peptides have long helix structures, which make them more likely to have the antimicrobial ability. At the same time, although the model predicts a long continuous helix structure for peptide $ID = 4$ and $ID = 9$, in fact, they have a small coil structure between two helix structures. It indicates that our model tends to predict a long continuous secondary structure instead of several discontinuous small fragments.

| ID | Sequence | Secondary Structure | C | H | uH |
|---|---|---|---|---|---|
| 1 | FLPLVRVWAKLI | –HHHHHHHHHH | 2.0 | 0.471 | 0.723 |
| 2 | FLSTVPYVAFKVVPTLFCPIAKTC | –HHHHHHHHHHHHHHHHHHHHHT– | 2.0 | 0.446 | 1.812 |
| 3 | FFGVLARGIKSVVKHVMGLLMG | –HHHHHHHHHHHHHHHHHHH– | 3.0 | 0.420 | 0.549 |
| 4 | GVLPAFKQYLPGIMKIIVKF | –HHHHHHHHHHHHHH— | 3.0 | 0.419 | 0.523 |
| 5 | VFTLLGAIIHHLGNFVKRFSHVF | -HHHHHHHHHHHHHHHHHHHHH– | 2.0 | 0.416 | 0.514 |
| 6 | FVPGLIKAAVGIGYTIFCKISKACYQ | –HHHHHHHHHHHHHHHHHHHHHT—- | 3.0 | 0.394 | 1.815 |
| 7 | ALWCQMLTGIGKLAGKA | –HHHHHHHHHHHHHHH | 2.0 | 0.344 | 0.506 |
| 8 | LLTRIIVGAISAVTSLIKKS | –HHHHHHHHHHHHHHHH– | 3.0 | 0.334 | 0.531 |
| 9 | FLSVIKGVWAASLPKQFCAVTAKC | –HHHHHHHHHHHHHHHHHHHHHT– | 3.0 | 0.334 | 0.660 |
| 10 | FLNPIIKIATQILVTAIKCFLKKC | –HHHHHHHHHHHHHHHHHHHHHT– | 4.0 | 0.334 | 1.940 |

Table 5: Ten generated peptides and their physical attributes and predicted structures. 'H' is the $\alpha$-helix, 'T' is the Turn and '-' is the coil.

## A.3 STRUCTURE CONDITION

As described above, controlling the secondary structure can affect the properties of generated peptides. Thus we limit the percentage of the coil structure with different ratio and calculate the physical attributes of generated peptides. The results are shown in Figure 6. We can find that with the decrease of number of coil structures, the percentage of positive peptides keep growing. However, for

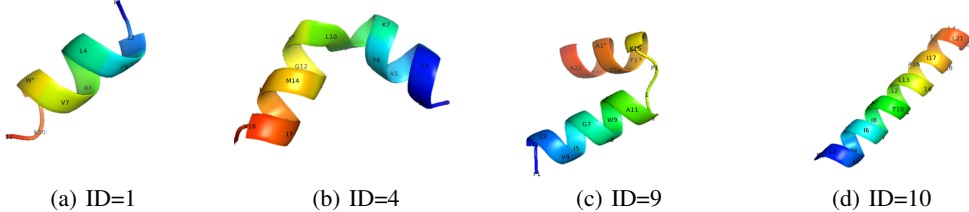

| (a) ID=1 | (b) ID=4 | (c) ID=9 | (d) ID=10 |

Figure 5: 3D structures for sequence **FLPLVRVWAKLI**, **GVLPAFKQYLPGIMKIIVKF**, **FLSVIKGVWAASLPKQFCAVTAKC**, **FLNPIIKIATQILVTAIKCFLKKC**. ID corresponds to Table 5.

|  | Uniq | C | H | uH | Combination |
|---|---|---|---|---|---|
| **APD** | 3222 | 68.75% | 27.96% | 4.72% | 6.15% |
| **Decoy** | 2020 | 21.83% | 8.81% | 1.98% | 0.10% |
| **Random** $p = 0.1$ | 2055 | 73.54% $\pm$ 0.69% | 37.93% $\pm$ 0.44% | 27.82% $\pm$ 0.31% | 8.74% $\pm$ 0.41% |
| **Random** $p = 0.2$ | 1831 | 69.00% $\pm$ 0.31% | 34.39% $\pm$ 0.31% | 22.70% $\pm$ 1.06% | 4.66% $\pm$ 0.66% |
| **VAE** | 475 | 56.99% $\pm$ 2.92% | 24.05% $\pm$ 3.28% | 9.82% $\pm$ 1.64% | 0.63% $\pm$ 0.74% |
| **AMP-GAN** | 1966 | **90.86**% $\pm$ 0.50% | 19.55% $\pm$ 0.34% | 21.26% $\pm$ 0.53% | 2.10% $\pm$ 0.35% |
| **PepCVAE** | 208 | 62.76% $\pm$ 1.58% | 12.61% $\pm$ 1.61% | 12.66% $\pm$ 2.80% | 6.68% $\pm$ 1.82% |
| **MLPeptide** | 2106 | 84.11% $\pm$ 0.39% | 11.02% $\pm$ 0.57% | **45.80**% $\pm$ 1.22% | 4.34% $\pm$ 0.38% |
| **LSSAMP** | 4886 | 75.06% $\pm$ 0.37% | **39.51**% $\pm$ 0.64% | 27.18% $\pm$ 0.41% | **9.96**% $\pm$ 0.07% |

Table 6: Physical attributes of sequences filtered by secondary structures. The notations are the same with Table 1.

hydrophobicity and hydrophobic moment, the percentage drop after 0.3. Therefore, we limit the length of the coil structure to 30% in our main experiments.

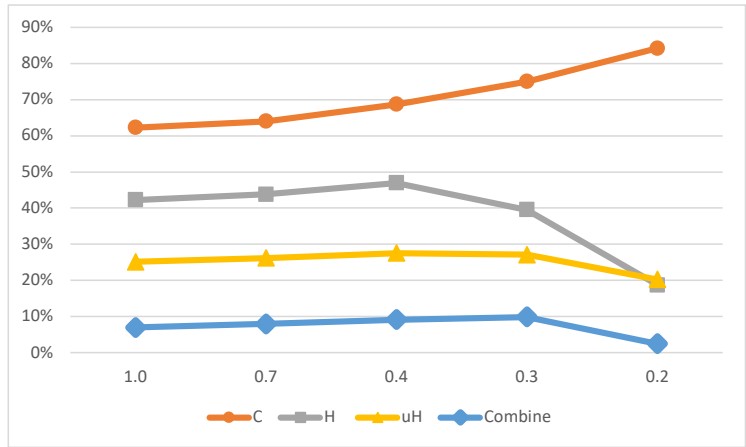

Figure 6: The physical attributes of peptides with different percentage of the coil structure. The x-axis is the maximum percentage and y-axis is the percentage of peptides that meet the attribute range.

## A.4 FILTER STRATEGY

Different from the existing work, we control the secondary structures during the generation phase instead of filtering sequences after the generation. But does this filtering mechanism make sense for other methods as the post-process? To answer this question, we use the same rules to filter the generated peptides of comparison partners. The results are shown in Table 6. Comparing Table 1

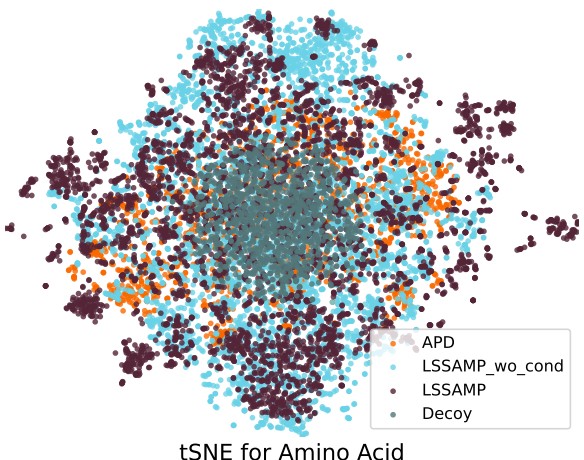

tSNE for Amino Acid

Figure 7: The tSNE plot for the distribution of amino acid in each sequence on four datasets.

and 6, we can find that all results are improved by limited sequences to the alpha-helical structures. It implies that by controlling the structure, the physical attributes can be improved, which further verifies the importance of secondary structures. However, the sequence number have significantly declined, indicating that this generate-then-filter pipeline is inefficient.

## A.5 DISTRIBUTION ON AMINO ACIDS

To illustrate the distribution of amino acids in generated peptides, we use the tSNE plot shown in Figure 7. We convert use the vector with each dimension representing the probability of a certain amino acid to represent the peptide. Then we use tSNE to convert the high-dimensional vector to 2D and visualize them. From Figure 7, we can obverse that there is a large overlap between LSSAMP w/o condition and APD, which indicates that our model has captured the global distribution of APD instead of collapsing to a local mode. Furthermore, LSSAMP covers APD and has some outliers. It implies that with the secondary structure condition, our model can not only learn the existing AMP space but also explore more possible space.

## A.6 NOVELTY

To measure the novelty of generated peptides, we define three evaluation metrics: Uniqueness, Diversity, and Similarity. **Uniqueness** is the percentage of the unique peptides. **Diversity** measures the similarity among generated peptides. We calculate the Levenshtein distance between every two sequences and normalize it by the sequence length. Then we average the normalized distance to get the mean as its diversity. The higher the diversity, the more dissimilar the generated peptides are. **Similarity** is the similarity between the generated peptides and the training AMP set (APD). For each generated sequence, we find a peptide from the training set which has the smallest Levenshtein distance with it and normalize the distance by its length. And we calculate the average length as the similarity.

From Table 7, we obverse that VAE has the highest diversity and lowest similarity. However, in Table 1 and 2, VAE performs the worst, which implies we can not measure the generation quality based on the novelty alone. Besides, the limitation on secondary structures will cause a significant decline in diversity. However, it does not result in the increase of the number of redundant peptides since the uniqueness does not increase. It implies that the restrictions make the model capture similar local patterns but not generate the exact same sequence.

|  | Uniqueness ↑ | Diversity ↑ | Similarity ↓ |
|---|---|---|---|
| **Random** $p = 0.1$ | 99.54% ± 0.02% | 0.871 ± 0.021 | 0.078 ± 0.001 |
| **Random** $p = 0.2$ | **99.99**% ± 0.02% | 0.971 ± 0.022 | 0.160 ± 0.001 |
| **VAE** | 98.60% ± 0.05% | **1.011** ± 0.038 | **0.584** ± 0.002 |
| **AMP-GAN** | 99.54% ± 0.07% | 0.907 ± 0.023 | 0.565 ± 0.007 |
| **PepCVAE** | 26.50% ± 0.58% | 0.367 ± 0.007 | 0.423 ± 0.005 |
| **MLPeptide** | 90.02% ± 0.30% | 0.850 ± 0.016 | 0.416 ± 0.010 |
| **LSSAMP** | 98.10% ± 0.09% | 0.664 ± 0.018 | 0.528 ± 0.005 |
| **LSSAMP w/o condition** | 97.64% ± 0.24% | 0.828 ± 0.013 | 0.580 ± 0.008 |

Table 7: The novelty of generated sequences.

| Codebook | PPL ↓ | Loss ↓ | AA Acc.↑ | SS Acc.↑ |
|---|---|---|---|---|
| [1] | 19.04 ± 2.84 | 2.94 ± 0.14 | 65.49 ± 3.49 | 83.41 ± 2.34 |
| [1, 2] | 3.84 ± 0.09 | 1.35 ± 0.02 | 99.40 ± 0.45 | 85.39 ± 0.26 |
| [1, 2, 4] | 3.32 ± 0.03 | 1.20 ± 0.01 | **100.00** ± 0.00 | 85.95 ± 0.42 |
| [1, 2, 4, 8] | **3.24** ± 0.16 | **1.17** ± 0.05 | 99.79 ± 0.20 | **87.20** ± 0.62 |

Table 8: The mean and standard deviation of results on codebooks. The meanings of symbols are the same as Table 4.

## A.7 REPRODUCTION

We describe the architecture of our model and the hyperparameters used in the experiments in detail in Section 4.1. We run the model for several times and calculate the mean and variance. In Table 8, we add the mean and the standard deviation of each setting in Table 4. We will also release our code after the anonymous period.

