# OpenReview forum: "Generating Antimicrobial Peptides from Latent Secondary Structure Space"
_ICLR.cc/2022/Conference — ICLR 2022 Submitted_

### Official Review · Reviewer_n6Pg · 2021-11-01

**Correctness:** 2
**Technical Novelty And Significance:** 2
**Empirical Novelty And Significance:** 1
**Recommendation:** 1
**Confidence:** 4

**Main Review:**

Unfortunately, I think this work suffers from a major conceptual flaw, namely, that there is not satisfactory gold standard to evaluate the utility of the proposed method.  The task being performed involves training a model from a set of known AMPs, simulating novel AMPs, and trying to ascertain whether these simulated AMPs are "good" in some sense.  The manuscript does this by asking whether the sequences capture some of the known properties of AMPs and whether they get assigned high scores by existing AMP classifiers.  This is not a useful mode of evaluation, for two reasons.  First, if these are truly the right performance measures to use, then it would make sense to simply optimize the model w.r.t. those measures directly.  Second, it is certainly plausible that a synthetic AMP sequence could do well according to these measures yet not have AMP function.  In practice, the only way to know whether an AMP simulator works is to synthesize that peptides and test them.  Short of that, I do not think it's possible to evaluate the accuracy of this type of model.

Another problematic aspect of this work is the use of deep models for modeling very small datasets.  The paper mentions that the AMP dataset is small and that some additional data is used for training.  I would like to see specifics about how large these datasets are and how they compare in size to the number of parameters in the model.

An aspect of evaluation that is missing here is the degree of sequence similarity to any member of the training set.  In principle, it should be easier to generate functional AMPs if we don't require that they differ much from known AMPs.  For example, if I randomly substitute a single amino acid in a known AMP, then the resulting "simulated" AMP is very likely to be functional.  Thus, the "random replacement" baseline method would be expected to do very well when p is small.  In practice, a good objective should probably include a term that captures the degree of novelty, by e.g. weighting sequences according to their nearest neighbor in the training set.

Minor:

Abstract: "few studies consider structure" Clarify in this sentence, rather than several sentences later, that this refers to secondary structure of the peptide.

On p. 2, I do not understand what "the conditioning vector" refers to.

Write out "VQ-VAE" when it's first used, and similarly for other abbreviations such as "FFN."

I think the footnote on p. 4 should be incorporated into the main text, and the meanings of the eight letters should be mentioned.

What does it mean to "kill bacteria in a physical way"? It's hard to
imagine any other way to kill them.

It is not clear to me exactly how the parameter p in the random baseline is interpreted: is this probability computed independently for each amino acid in the peptide?

"multiply ideal" -> "multiple ideal"

"proprieties" -> "properties"



**Summary Of The Paper:**

This manuscript addresses the problem of generating novel antimicrobial peptides (AMPs).  The approach uses a deep learning framework, and the primary novelty is to include in the model information about peptide secondary structure.  The hypothesis that such information could improve the ability to capture valuable information about AMP function is plausible a priori.


**Summary Of The Review:**

This paper addresses a problem for which no good gold standard is available, short of experimental synthesis and testing of the proposed sequences.

---

> ### Author Response · Authors · 2021-11-23
> **Thanks for your valuable comments!**
>
> Thanks so much for your comments and constructive suggestions! We will first address your concerns about the evaluation metric and then answer other questions. Hope our replies can solve your concerns and any futher comments are welcome!
>
> ---
>
> **Q1: There is not satisfactory gold standard to evaluate the proposed method.**.
> A: This is an important problem for computation-based and AI-based methods in drug design. The only reliable way to verify the generated molecules or proteins is the wet experiments. However, we aim to **accelerate** the drug discovery process by generating sequences that are **more likely to be AMP**. Besides, we hope that the model can find some new patterns of AMP sequences that were ignored before. So we employed two evaluation metrics, one is based on empirical rules and the other is classifiers. We also designed some wet experiments to test some of the sequences generated by our model. But it is impossible to examine all sequences generated by the model because the wet experiment is **expensive** and **time-consuming**.  Since we can hardly use the wet experiment results to optimize our model, we introduce these computational metrics to improve and evaluate our model in a faster way. *At the same time, we have submitted several top results to the wet laboratory to examine the performance, but the results have not been available for the long and complex process.*
> We will also commit to developing a more accurate metric, which is one of the main problems in CADD(computer-aid drug design) and AADD(AI-aid drug design).
>
> To generate peptides that are more likely to be AMPs, we use physical properties and public AMP classifiers, both of which are widely used in previous works (Capecchi et al., 2021; Das et al., 2020; Van Oort et al., 2021).
>
> (1) These **physical properties** are widely thought to have an important effect on the antimicrobial properties of peptides (Hancock  &
> Rozek,
> 2002;  Boman, 2003; Aronica et al., 2021). These properties only depend on the amino acid sequences (see Section 4.2.1). Therefore, from learning the sequence features from existing AMPs, we suppose the model can capture these characteristics and the experimental results verify its effectiveness. Besides, the function of a peptide is determined by its amino acid composition and structure. So we model the sequence and structure at the same time to control the ideal properties. The results in Table 1 show that by introducing the structures, our model can implicitly optimize towards these measures and generate sequences satisfying these properties.
>
> (2) The **public AMP classifiers** are also often used in AMP generation, but it is also untrustworthy for their generalization ability. To reduce the bias of the single classifier, we apply seven public classifiers and use the average score to measure the performance.

---

> > ### Author Response · Authors · 2021-11-23
> > **Some other concerns**
> >
> > **Q2: how large these datasets are and the number of parameters**.
> > A: As stated in 'Dataset' in Section 4.1, we use proteins from UniProt with the limitation of length 100 D_{r} to pretrain our model. The number of examples is 57k. The other dataset used for secondary structure prediction D_{s} has 46k examples. The number of parameters of LSSAMP is about 1.8M. We think our dataset is enough to train our model.
> >
> > **Q3: what "the conditioning vector" refers to**.
> > A: AMPGAN uses a vector to control the generated peptides. It is combined with the latent representation and fed to the generator to generate new peptides. The vector is composed of the target microbes, the target mechanisms, the MIC50(activity value) and the sequence length. The detailed information can be found in Van Oort et al., 2021. This description seems a little confusing in the introduction and we have modified the description of the existing work.
> >
> > **Q4:  What does it mean to "kill bacteria in a physical way"?**.
> > A: Traditionally, there are two ways to kill bacteria. One is **physical** and the other is **chemical** methods. *Heating, X-ray,  ultraviolet rays and so on* are to kill bacteria in a physical way. On the other hand, chemical methods usually use a special chemical material to react with the bacteria to kill them, such as *antibiotics, ethylene oxide, nitrogen dioxide, ozone, hydrogen peroxide and so on*.
> > The **POINT** we focus on is that the most clinically used sterilization method, chemical antibiotics, will result in resistance to the rapid evolution of bacteria. Antibiotics usually interfere with bacterial replication by binding important proteins, which is easy to resist by changing the structure of the protein or producing a new enzyme to break down the antibiotics. So we focus on designing AMPs targeting the lipid membrane that kill bacteria by breaking the membrane. This mechanism guarantees that it is hard to resist, for it is unlikely that bacteria can live without its membrane. So here we call it "physical way" because AMPs make some holes in the bacteria membrane.
> >
> > **Q5: how the parameter p in the random baseline**.
> > A: Each amino acid has the probability p to mutate into another amino acid and this mutation is independent.

---

> > ### Comment · Reviewer_n6Pg · 2021-11-29
> > **Lack of gold standard**
> >
> > I understand that the goal of the work is to accelerate the drug discovery process, but my concern is that the method proposed here lacks a gold standard for validation.  The only way to do this, if you really aim to generate novel AMPs, is to test them in the lab.  The authors say they have done this but the results are not available yet. I think they should wait until the results come in and then submit the paper, since without those results we don't have any convincing way to know whether the methods works well.

---

> > > ### Author Response · Authors · 2021-11-30
> > > **Thanks for the further discussion!**
> > >
> > > We are sorry for the unavailable wet-experimental results currently and will add them as soon as possible. However, we still insist that the current comparisons between different models make sense from both AMP mechanism theories and common practices in related works. It indicates we can provide a more effective and efficient generative model. The usage of computational metrics without the wet-experimental results to measure performances of the neural network model is also widely-accepted practice in other drug discovery tasks, such as molecule generation[1-5].
> > > Any further discussion is welcome.
> > >
> > > ---
> > >
> > > [1] Junction tree variational autoencoder for molecular graph generation, ICML 2018.
> > > [2] Graph convolutional policy network for goal-directed molecular graph generation, NeurIPS 2018.
> > > [3] Hierarchical generation of molecular graphs using structural motifs, ICML 2020.
> > > [4] Augmenting genetic algorithms with deep neural networks for exploring the chemical space, ICLR 2020.
> > > [5] MARS: Markov Molecular Sampling for Multi-objective Drug Discovery, ICLR 2021

---

### Official Review · Reviewer_bThw · 2021-11-02

**Correctness:** 3
**Technical Novelty And Significance:** 3
**Empirical Novelty And Significance:** 2
**Recommendation:** 6
**Confidence:** 4

**Main Review:**

I think the idea of utilising secondary structure to aid AMP generation is useful, which could be seen from the improved performance from LSSAMP w/o to LSSAMP in Table 2. This result suggests that secondary structure is an important factor to consider in AMP design. The physical properties of generated peptides (table 1) are also impressive.

I feel the paper is on the boarder line due to the following concerns.
1. In table 2, the performance is generally worse than MLPeptide, which suggests that the proposed method have space to improve.
2. Are secondary structure really important? Could you use pep-fold (https://bioserv.rpbs.univ-paris-diderot.fr/services/PEP-FOLD3/) to get the secondary structure of peptides generated by other methods such as Random, VAE, AMP-GAN, MLPeptide, then filter out peptide sequences as LSSAMP. This will help to answer the improvement is universal, rather than unknown modelling glitches in LSSAMP.
3. Page 6, right before sec 4.2, The batch size for Dr (n=57k) and Ds (n=46k) are 30,000 and 10,000, which are very large. What is the consideration here?
4.  Table 1, the combination score of LSSAMP is much higher than that of APD, which is suspicious. In theory APD performance should be the upper bound of LSSAMP. Is this suggesting LSSAMP collapse to a local mode, i.e. generated pep sequences mimicking a subset of APD? Page 13, supplementary A2, all structures are long stretches of H. I suggest the authors convert each pep sequence to a vector with each dimension representing the probability of a certain amino acid of the given pep sequence. Then make a tSNE plot of generated pep sequences together with APD dataset, this will help to clarify the diversity of generated pep sequences.
5. Page 5, prior model: I think the dependency is already modelled by the encoder, what is the motivation of  imposing a further layer of dependency on the hidden layer?
6. It will be nice if a section could be added regarding the reproducibility of experiments in the paper.

Minors
1. Page 2, “Experimental results show that LSSAMP can …. with multiply”: multiply to multiple
2. Page 5, sec 3.4, paragraph 2, “a peptide dataset Dr” to “ Ds”


**Summary Of The Paper:**

Generating antimicrobial peptide (AMP) is a challenging task. This paper assumes a latent representation is both associated with the amino acid sequence and its secondary structure of peptide. This paper adopts a VQ-VAE network to learn the latent representation of a given peptide sequence, as well as its secondary structure. Then a transformer-based language model is used to model the prior distribution of the latent representations. Random peptide sequences can be generated at this point. The paper evaluates the physical properties of the random peptide sequences, and compared the proportions of these random peptide sequences that are classified as AMP. The comparison shows that secondary structure is important for the generation and the quality of the generated peptide sequences is high.


**Summary Of The Review:**

I think this paper is on the boarder line. I tend to accept it if the secondary structure information could be proven to be useful also for other methods.

---

> ### Author Response · Authors · 2021-11-23
> **Thanks for your valuable comments!**
>
> Thanks so much for your comments and constructive suggestions! We will address your concerns respectively in the following paragraphs. Hope these replies could resolve your concerns, and any further comments are welcome!
>
> ---
>
>
> **Q1: Are secondary structure really important?**.
> A: Please refer to the general response Q5. Besides, We use the same rules to filter the generated peptides of comparison partners. The results are shown in Appendix A.4. Comparing Table 1 and 6, we can find that all results are improved by filtering sequences with secondary structures. It implies that by controlling the structure, the physical attributes can be improved, which further verifies the importance of secondary structures.
>
> **Q2:  batch size 30,000 and 10,000**.
> A: We set the batch size as the maximum token in a batch, so 30,000 and 10,000 are the maximum token numbers. The large number makes the training more robust and faster and can make full use of the GPU memory. We have modified the description to make it clear.
>
> **Q3:  the combination score of LSSAMP is much higher than that of APD**.
> A: As stated in Section 4.3, LSSAMP uses extra structural control (< 30% coil and > 4 alpha-helix ) to improve the alpha-helical structures in generated peptides. The physical attributes have been improved, which corresponds to our theoretical basis that the amino acid sequence and its structure will determine its properties.  Also, from Table 1, we can find LSSAMP w/o condition has a similar score as APD, which implies that our model has captured the distribution of APD and this can be further improved by the modification on secondary structures.
>
>
> **Q4: a tSNE plot of generated pep sequences on amino acids**.
> A: In  Figure 1, we plot the distribution of amino acids in the first subfigure via the histogram, which shows the percentage of each amino acid type for each model. We also add the tSNE figure as the reviewer suggested in Appendix. Figure 7 shows that our model can capture the global characteristic of APD instead of collapsing to a local mode.
>
>
> **Q5: prior model**.
> A: According to the VQVAE(van den Oord et al., 2017), the prior distribution over the discrete latent is a  categorical distribution. During training,  the dependency of the discrete latent is modeled by the encoder. However, during sampling, we do not use the encoder and directly sample on the codebook by generating a sequence of embedding indexes on the codebook. Thus, to model the dependency between the index sequence, we train the autoregressive language model to generate the index sequence.
>
> **Q6: Reproducibility**.
> A: We have added a reproduction section in Appendix. We also conduct the main experiment repeatedly and calculate the mean and variance for each model. We will also release our code after the anonymous period.

---

### Official Review · Reviewer_s64x · 2021-11-09

**Correctness:** 3
**Technical Novelty And Significance:** 3
**Empirical Novelty And Significance:** 2
**Recommendation:** 5
**Confidence:** 4

**Main Review:**

I believe that pretraining the model on a larger dataset, and the usage of both the aa sequence and secondary structure, are items that contribute to the strength of the paper. More generally, the ablation study Table 3 is very interesting in order to understand better the performance of the proposed model. Additionally, the model seems to be well-thought to fit the sequences under study.

However, the work of the authors show some weaknesses, which are described below.

1. Some points are unclear and could be better explained. For example, the sizes of the datasets of LSSAMP and LSSAMP without condition Table 1 are very similar, did you resample between the two or did you remove peptides that do not verify the condition? Also, I am not sure how the AA Acc and SS Acc were computed, Table 3. Could you give a bit more details?

2. We could question the comparison partners. For example, the authors do not compare to PepCVAE (Das et al 2018), while mentioning it at the beginning of the manuscript. Does PepCVAE correspond to the VAE proposed Table 2? Also, while the model of the author uses pretraining, I am not sure to understand whether the proposed comparison partners do. I believe that it could be fair to use pretraining for at least a few of the comparison partners in order to have a fairer comparison, especially because pretraining in this application is not a new idea (see Das et al 2018).

3. The results of the proposed approach are not entirely convincing. First, the proposed model seems to underperform compared to MLPeptide (Table 2). Second, it seems that the author have to filter, after generation, peptides with more than 30% coil structure or helix structure lengths shorter than 4. Given the simplicity of these rules, it seems surprising that the model is not able to largely capture them, if they are largely present in the training set. Does this filtering also improve the results of the comparison partners? If yes, by how much? And if no, does it mean the comparison partners better capture these simple rules? Third, in the APD dataset, very few peptides (~6%) seem to verify the boundary conditions defined for the three C, H, and uH attributes. Therefore, I am wondering if it is reasonable to evaluate the generated sequences based on the defined ranges for these attributes ("combination" Table 1), or based on these attributes at all. Fourth, in the paragraph Codebook Number, the authors claim that the reconstruction accuracy achieves the best performance when the codebook number is 3. To me, it seems that the AA Acc performance could be indistinguishable between 2, 3 or 4 codebooks as the numbers are very close. Could you provide standard deviations to be able to better understand the results?

4. While the authors analyse the similarity between the generated sequences and the real ones, the authors do not mention possible overfitting (how similar the generated sequences are to the real ones) or mode collapse (how similar the generated sequences are between themselves). In particular, it seems that the generated secondary structure is not diverse as we can see Table 5 in the appendix. I believe that these two points (similarity and diversity) could be interesting to study to have a better understanding of the performance of the proposed approach.



Typos:

several times 'mutliply' is used instead of 'multiple'

Section 3.1: to reconstruct x -> x should be bold

Section 3.1: j \in K -> j \in {1, ..., K}

Section 3.4: the second D_r should be D_s

Section 3.4: the sum (large sigma signs) should indicate the letter on which the sum is done.

Section 4.1: will be padding -> will be padded

Section 4.1: codebook -> codebooks

Section 4.3: 6.27% should be 6.15%? (to match Table 1)

There are a few other typos, I think it could be worth rereading carefully the manuscript.


Unclear sentences/plots:

Section 3.4: "the number that the embedding ... via Eqn.1 " -> maybe it should be " the number of times that the embedding vector e_k is ..."

Table 3/4: the caption is very confusing as there is no space between Table 3 and 4.

The 3D plot Figure 3 is very hard to read as all the points are overlapping.




**Summary Of The Paper:**

The paper proposes a multi-scale VQ-VAE model to generate antimicrobial peptides (AMPs). The model both train on the amino-acid (aa) sequences and on secondary structure. The author evaluate the model by comparing attributes of the generated sequences with real AMPs. The authors also perform an ablation study of their model, an extensive benchmark of existing generative models that have been used for peptides.

The main contributions of the authors are:
- Developing a generative model that accounts for both the aa sequences as well as the secondary structure of the AMPs.
- A large set of experiments to validate their approach, comprising the study of attributes of the generated sequences, benchmark of existing methods, evaluation of the generated sequences with existing predictors and an ablation study.



**Summary Of The Review:**

To summarise, I believe that the paper is below the acceptance threshold as:
- the proposed method do not seem to beat the state of the art (MLPeptide)
- some analyses of the generated sequences are missing, such as similarity study between the generated sequences and the real ones, and diversity of the generated sequences.
- the value of some validation experiments (for example Table 1) or of the filtering step is unclear.

I could increase the score if the authors address these points.

---

> ### Author Response · Authors · 2021-11-23
> **Thanks for your valuable comments!**
>
> Thanks so much for your comments and constructive suggestions! We will address your concerns respectively in the following paragraphs. Hope these replies could resolve your concerns, and any further comments are welcome!
>
> ---
>
>
> **Q1: LSSAMP and LSSAMP without condition**.
> A: The 'Size' in Table 1 indicates the unique sequences from the generated 5000 sequences. We have changed it to 'Uniq'. We resample LSSAMP instead of just removing sequences that do not satisfy the condition from LSSAMP w/o condition.
>
>
> **Q2: AA Acc and SS Acc**.
> A: The accuracy can be calculated by Count(pred==gold)/Count(gold), where pred is the model-predicted tokens and the gold is the ground-truth ones. For AA accuracy, the token is the amino acid and for SS accuracy the token is the secondary structure type of each amino acid.
>
> **Q3: PepCVAE & VAE & MLPeptide**.
> A: Please refer to the general response Q2 and Q3.
>
> **Q4: Pretraining in comparison partners**.
> A: Pretraining is not a new idea for this task since the known AMP dataset is really small. Among the proposed comparison partners, VAE and AMP-GAN both use the unsupervised UniProt dataset for training. As described in the baseline of Section 4.1 , VAE is pretrained in the same D_{r} (46k) with LSSAMP for comparison. AMP-GAN is trained on 49k negative sequences from UniProt and 7k positive AMPs.
>
>
> **Q5: Does this filtering also improve the results of the comparison partners?**.
> A: In fact, we do not resample after generation. We use the conditions of the secondary structure during the generation phase instead of filtering them after the generation, which is different from current methods. But does this filtering mechanism make sense for other methods as the post-process?
> To answer this question, we use the same rules to filter the generated peptides of comparison partners. The results are shown in Appendix A.4. Comparing Table 1 and 6, we can find that all results are improved by filtering sequences with secondary structures. It implies that by controlling the structure, the physical attributes can be improved, which further verifies the importance of secondary structures. However, the unique sequence number has significantly declined, which indicates that the generate-and-filter pipeline is inefficient.
>
>
> **Q6: very few peptides (~6%) in APD dataset meet three attributes**.
> A: We plot the distributions of C, H and uH on APD in Appendix A.1. From Figure 4, we can find that the range of the distribution is large, which may be due to various target microbes and mechanisms of these AMPs. Thus, in order to find a suitable range, we create a decoy dataset and choose the range where the fraction of APD exceeds decoy ones. From Table 1, we can find that only 0.1% of decoy ones meet the constraint. Since our goal is to find peptides that are more likely to be AMPs (precision) instead of finding all AMPs (recall), so this stricter scope is more preferable.
>
>
> **Q7: indistinguishable between 2, 3 or 4 codebooks as the numbers are very close**.
> A: We conduct this experiment several times and calculate the standard deviation for it. The results are added in Table 8 in Appendix. We can obverse that although the gaps between different setting is not very large, the trend is always the same.
>
>
>
> **Q8: the generated secondary structure is not diverse**.
> A: It is because we control the secondary structure to alpha-helix as stated in Section 4.3, since most known AMPs have the alpha-helical structure. The fragment of alpha-helix is an important characteristic for AMPs and is often used to measure and select generated peptides. The detailed explanation can refer to general response Q5.

---

> > ### Comment · Reviewer_s64x · 2021-11-29
> > **Thanks for the detailed answers.**
> >
> > I would like to thank the authors for providing clear answers to the questions that were asked and for the clarification concerning the filtering step. The new analyses seem to reinforce the importance of secondary structure more clearly. However, I still keep two concerns that were mentionned during the first review:
> > - MLPeptide is still better than the proposed model according to Table 2 and two criteria out of 3 Table 7.
> > - The value of the analysis of the physical attribute (Table 1 and 6) is not clear in my opinion. If only 6.15% of the AMPs have the combination, then the analysis leaves aside >93% of potential AMPs of interest. I understand that one is not interested in generating all possible AMPs but I believe that being interested in generating AMPs among a group that represent 6% of the whole space is very restrictive.
> >
> > Therefore, I am not willing to increase the score above the acceptance threshold.

---

> > > ### Author Response · Authors · 2021-11-30
> > > **Thanks for the further discussion!**
> > >
> > > The antimicrobial mechanisms are various for different target microbes and by limiting the range of our attributes, we only focus on a specific one. The three physical attributes are also widely used in related works (Capecchi et al., 2021; Das et al., 2020; Van Oort et al., 2021). Besides, the classifiers are less reliable than physical attributes and we think the outperformance on the combinations of physical metrics can make up for them. Still appreciate your valuable comments!

---

### Author Response · Authors · 2021-11-23
**Response to all the reviewers**

We would like to express our sincere appreciation to the reviewers and the area chair for your efforts! We add some experiments and modify our paper based on these considerations. The main modifications are as follows:
1. Add a new baseline PepCVAE. The results are shown in Table 1 and 2.
2. Calculate the mean and standard deviation for each model in Table 1 and 4 (the full Table 8 is put in Appendix) to make the results more convincing.
3. Add Novelty, Diversity, and Uniqueness to evaluate the models, which are also put in Appendix (Table 7).
4. Add the tSNE analysis (Figure 7) for the distribution of amino acids in Appendix.
5. Add experiments to explore how the restrictions of secondary structures affect other models. The results are shown in Table 6 in Appendix.
6. Polish the description of Introduction Section.
7. Correct the typos and other errors.

##########################################################################


Here, we solve several common concerns.
**Q1: Diversity and Novelty**.
A: We define Uniqueness, Diversity and Similarity to measure the novelty of generated sequences. **Uniqueness** is the percentage of the unique peptides. **Diversity** measures the similarity among generated peptides. **Similarity** is the similarity between the generated peptides and the training AMP set (APD). We obverse that VAE has the highest diversity and lowest similarity, but it performs the worst in physical attributes and classifiers. Besides, the limitation on secondary structures causes a significant decline in diversity and does not decrease the uniqueness, which implies the restrictions make the model capture similar local patterns. The detailed information is put in the Appendix.


**Q2: PepCVAE & VAE**.
A: Since the original paper did not release its code, we have reproduced the model from https://github.com/wiseodd/controlled-text-generation and added the results in our paper (Table 2 & 3 & 6 & 7) . The implementation details can be in Section 4.1. Compared with other models, PepCVAE tends to generate redundant sequences, which results in a significant decline in the number of unique sequences. Besides, for AMPMIC and IAMPE, PepCVAE has a significant advantage over other methods, but it has a poor performance in other classifiers.


**Q3: Underperform compared to MLPeptide**.
A: For the two evaluation metrics we use to evaluate models, our model outperforms MLPeptide in physical attributes by a large margin (6.74%) and achieves comparable performance in classifiers. The physical attributes are based on empirical rules and the theories of antimicrobial mechanisms, that are more general and explanatory. The AMP classifier highly depends on its training domain and is thought less reliable. From Table 2, we can find that the results of AMP classifiers vary from each other. For example, our model achieves the best results in SVM and amPEP, and MLPeptide outperforms others in RF and Scanner. However, PepCVAE is the greatest for DA, AMPMIC and IAMPE. We think that the comparable results on classifiers and the significant improvement on physical attributes can verify the effectiveness of our model.

**Q4: Typo and grammar**.
A: Thanks for all comments for typos and other errors. We have corrected all details pointed out and rechecked our paper carefully.

**Q5: About the necessity of secondary structure**.
A: As we describe in Section 1, the secondary structure is very important in the antimicrobial mechanism of AMPs and most known AMPs have alpha-helical structures. In fact, current works often include the discussion on the alpha-helix of generated sequences to evaluate their effectiveness. If the generated peptide has the alpha-helix structure as most known AMPs, it has a higher probability to have antimicrobial activity.

Das et al. (2018), Van Oort et al.(2020), Dean & Walper (2020), and Das et al. (2021) model the structures of top-k generated peptides. If these peptides have an alpha-helical structure, it means they are more likely to be AMPs, which can be a metric to show the effectiveness of their generative model. Capecchi et al. (2021) use this metric ( a predicted helicity fraction > 80% ) to filter the generated sequences.

Since the alpha-helical structure is necessary for AMPs, we use this restriction for the generation. As stated in Section 4.3, we limit the generated peptides with at least 4 alpha-helix and no more than 30% coil structure. That is why the cases shown in Appendix A.2 all have a long alpha-helical structure.

---

### Decision · Program_Chairs · 2022-01-20

**Decision:**

Reject

**Comment:**

Reviewers agreed that taking into account the secondary structure in addition to the amino acid sequence, although not new in bioinformatics, may be a good idea in the context of deep generative models of peptides. On the other hand, all reviewers also agreed that the experimental results do not allow concluding about the potential benefit of the method, i.e., whether it is likely to produce potent AMPs (and whether it does it better than existing methods). Indeed, the proposed computational criteria can not replace a proper experimental validation, and it is not clear whether a "better method" on the computational criteria will be "better" in the real world. Second, the results on the computational criteria are not convincing: regarding the physical properties, it remains debatable to claim that a method is good if it outputs many AMPs that fulfill the criterion, while less than 7% of the true AMPs do; and regarding the computational prediction of being an AMP, the proposed method is outperformed by existing ones. In conclusion, we consider that the paper is not ready for publication at ICLR, since there is no significant methodological novelty nor significant experimental results if this is an application paper, and we encourage the authors to consider a publication with wet lab experiments to demonstrate the relevance of the method.